# Optimized Protocol for Primary Rat Hepatocyte Isolation and a Model for Investigating Experimental Steatosis

**DOI:** 10.3390/mps8050111

**Published:** 2025-09-19

**Authors:** Amani A. Harb, Mohammad AlSalem, Shtaywy Abdalla

**Affiliations:** 1Department of Biological Sciences, School of Science, The University of Jordan, Amman 11942, Jordan; a.harb@ammanu.edu.jo; 2Department of Allied Sciences, Faculty of Arts and Sciences, Al-Ahliyya Amman University, Amman 19111, Jordan; 3Department of Anatomy and Histology, School of Medicine, The University of Jordan, Amman 11942, Jordan; m.alsalem@ju.edu.jo

**Keywords:** primary hepatocyte, rat liver, isolation, perfusion, culture, steatosis

## Abstract

Background: Primary hepatocytes are excellent models for studying liver functions and liver diseases. However, obtaining high yields of viable hepatocytes remains technically challenging, limiting their broader applications. Most conventional methods rely on a two-step collagenase perfusion technique. Despite its widespread use, this approach has several limitations that reduce the success rate of hepatocyte isolation and culture. The procedure involves multiple parameters that are continually being optimized in order to obtain hepatocytes in high yield and quality that can be used to provide insights into their physiology and pathophysiology. Aim: We aimed to enhance the success rate and reproducibility of hepatocyte isolation with high yield, enabling analysis of diverse physiological and pathophysiological aspects of lipid metabolism. It also establishes an in vitro steatosis model for evaluating therapeutic drugs and molecular interventions. Methods: Rat liver was perfused in situ with EDTA buffer followed by collagenase IV. Liver was then isolated, and hepatocytes were mechanically liberated, filtered, and purified through density-gradient centrifugation. Viable cells were cultured at 700,000 or 1 million cells/well for 24 h. The monolayer was incubated in lipogenic media for an additional 24 or 48 h. Hepatocytes were fixed, neutral lipids were stained using Oil Red O, and the stained area was quantified using Image J software version 1.54. Results: Yield of hepatocytes was ~75–90 million cells/liver, with viability of 86–93%. Cells seeded at 700,000 and 1 million cells/well reached confluences of 60% and 80%, respectively, after 24 h. Steatosis was then induced with lipid accumulation reaching 21% of image area after 24 h and 25% after 48 h. Conclusions: The current protocol presents an efficient and highly reproducible method for isolating primary rat hepatocytes in high yield with high viability. Additionally, the protocol provides a foundation for studying the pathophysiology of fatty liver disease.

## 1. Introduction

Primary hepatocytes are a key tool in biomedical research. They are considered indispensable for many applications, as they perform most hepatic functions, including lipid metabolism and drug metabolism. Their utilization as an in vitro model is common for studying various aspects of liver physiology and pathology [1,2,3,4]. Additionally, there is significant potential for hepatocyte organoids, derived from fetal or adult livers, to greatly influence regenerative medicine [5]. As such, the isolation and culturing of primary hepatocytes for in vitro studies are highly demanded for investigating liver physiology and to develop therapeutics for liver diseases.

Over the past decade, several strategies have been developed to enhance the isolation of viable primary hepatocytes and to improve their viability and functionality in culture. Most isolation techniques were based on the two-step collagenase perfusion method first introduced by Per Ottar Seglen in 1976 [6]. This process begins with an in situ perfusion of the liver using a Ca^2+^-free buffer to flush out blood and circulating cells and loosening cell-to-cell connections within the liver tissue. Subsequently, the liver is perfused with a buffer containing Ca^2+^ and collagenase to break down the extracellular matrix and release individual hepatocytes [7].

While this method is widely used, several factors can lead to reduced cell yield and viability. Various studies highlighted key issues compromising this process. For example, Shen et al. (2012) [2] reported bacterial and fungal contamination, perforation of the portal vein through which the perfusion initiated, improper temperature of perfusion buffers (below 37 °C), slow perfusion speeds for collagenase-containing buffer (<25 mL/min for rats), excessive collagenase digestion, and finally damage caused by harsh pipetting, as factors that can compromise cell viability. Salem et al. (2018) [8] identified additional challenges as common drawbacks in hepatocyte isolation, including improper cannulation, inadequate washing with Hank’s balanced salts solution (HBSS), improper collagenase concentration, prolonged perfusion time, and issues with buffer temperature and flow rate. Charni-Natan and Goldstein (2020) [9] recognized factors such as cannulation technique, air bubbles in tubing, centrifugation speed, and the use of small-pore serological pipettes (<25 mL) during cell transfer that also impacted hepatocyte viability. More recently, Poggel et al. (2022) [5] discussed how hypoxia and mechanical stress caused by stretching and tearing tissue with forceps during cell dissociation could further reduce cell viability during the conventional isolation procedure. These observations underscored the importance of optimizing key parameters in the hepatocyte isolation technique to maximize cell yield and to maintain high cell viability at the same time.

When exploring the details of the protocol, the isolation of viable primary hepatocytes is technically a challenging process that requires skilled, trained personnel and careful optimization of numerous parameters. Several published protocols for isolating hepatocytes from rat and mouse livers differed in key experimental conditions, such as perfusion buffer flow rate (4 mL/min to as high as 40 mL/min) [2,8,9,10,11,12], temperature (37 °C to as high as 45 °C) [2,8,9,10,11,12], anterograde perfusion via the portal vein [2,11,13] or retrograde via the inferior vena cava [8,9,12], Percoll concentration (36% to as high as 48%) [2,8,9,12,13], collagenase type (I, IA, or IV) and concentration (25 µg/mL to as high as 500 µg/mL), centrifugation speed (100 to 800× *g*) [2,8,9,10,12], and finally hepatocyte dissociation technique (e.g., cutting liver lobes into pieces, gentile pressing, shaking, perforation of liver lobes, massaging through sterile gauze) [2,8,9,10,11,12].

On the other hand, for hepatocytes culturing, some variability was also reported. Some studies used collagen coating for the culture dishes or wells to enhance hepatocytes attachment [9,10,14,15], while others showed that collagen coating may not be necessary [2,8,12]. These variations contribute to the difficulty in achieving consistent reproducibility of the technique.

The lack of reproducibility may result from the lack of clarity in describing critical steps. For a successful and reproducible isolation of primary hepatocytes, it is essential to encourage further refinement and improvement of hepatocyte isolation techniques especially when taking into consideration the high variability in the different components of the technique and whether each component is mandatory or optional. Therefore, the goal of this current protocol was to increase the success rate and reproducibility of the technique, and the harvesting of relatively a large number of cells from the same liver to be used for the different experiments. In this communication, we present a detailed description of the critical steps, describing some of troubleshooting problems, in order to improve the technique to help beginners acquire the skills necessary for successful perfusion, isolation, and culturing of hepatocytes, thus minimizing technical errors and ensuring high cell viability. Furthermore, this work provides an in vitro model for both mild and severe steatosis, enhancing the quality of lipid staining in primary rat hepatocytes and providing a platform for studying some of the receptors involved in lipid metabolism.

## 2. Experimental Design

### 2.1. Materials

Antibiotic-antimycotic solution (penicillin-streptomycin-amphotericin B, 100×, Servicebio, Wuhan, China; Cat. No. G4015)Bovine serum albumin (fraction V, fatty acid free, Oxford lab chem Co., Mumbai, Maharashtra, India; Cat. No. A-00395) or bovine serum albumin lyophilized containing ~1% fatty acid (Biowest, Nuaillé, France; Cat. No. P6154)Collagen Type I (Sigma-Aldrich, Saint Louis, MO, USA; Cat. No. C4243)Collagenase IV (Sigma-Alderich, Saint Louis, MO, USA; Cat. No. C4-22-1G)D-glucose (Fisher Chemical Co., Ltd., Guangzhou, China)DMEM low glucose (1×) (Cat no. ECM0060L, Euroclone, Pero, Italy)EDTA (Tokyo Chemical Industry Co., Ltd., Tokyo, Japan)FBS (Capricorn Scientific, Ebsdorfergrund, Germany)HBSS with Ca^2+^ and Mg^2+^, without phenol red (1×) (Cat. No. HBSS-1A, Capricorn Scientific, Ebsdorfergrund, Germany)HBSS without Ca^2+^, Mg^2+^, and phenol red (1×) (Cat. No. HBSS-2A, Capricorn Scientific, Ebsdorfergrund, Germany)HEPES (AppliChem, Darmstad, Germany)Insulin (Santa Cruz Biotechnology, Heidelberg, Germany; Cat. No. sc-360248)Isopropanol (assay 99.9%, Scharlab S.L., Barcelona, Spain)Ketamine (Alfasan, Woerden, Holland)N-acetyl- L-cysteine (≥99% TLC, Sigma-Aldrich, Saint Louis, MO, USA)Oil Red O stain (Serva Feinbiochemica, Heidelberg, Germany)Paraformaldehyde (Fluka, Buchs, Switzerland)Percoll (Sigma-Alderich, Saint Louis, MO, USA; Cat. No. P4937)PBS (1×) (Euroclone, Pero, Italy)Sodium oleate (purity 99%, Loba Chemie pvt. Ltd., Mumbai, India; Cat. No. 5958D)SpermGrad™ (Vitrolife, Gothenburg, Sweden; Cat. No. 10099)Trypan blue stain (Gibco, New York, NY, USA; Cat. No. 15250061)William’s medium E (1×) + GlutaMAX™ (Gibco, London, UK; Cat No. 32551-020)Xylazine (Alfasan, Woerden, Holland)

### 2.2. Equipment

A CO_2_ incubator (Bio San, Riga, Latvia)A heating magnetic stirrer plate (Velp Scientifica, Arec.x, Usmate, Italy)A laminar flow hood (Airtech VS-1300L-U, Tianjin, China)A microscope (Micros Austria, MCX1600, Veit/Glan, Austria)A peristaltic pump (Watson-Marlow 502S, Cornwall, England)A refrigerated benchtop centrifuge (MPW-260R, Medical supply company Ltd., Dublin, Irland)A sterile disposable infusion IV tube (1.5 m in length and 3 mm in diameter) with a 24 G cannulaA water bath (Grant Instruments, Royston, England)6-well-plates (NunclonTM Delta Surface, Thermo Scientific, Roskilde, Denmark)

## 3. Procedure

### 3.1. Perfusion Setup

The perfusion system consisted of a water bath pre-set to 40 °C, a peristaltic pump set to a flow rate of 10 mL/min, a sterile disposable infusion IV tube with a 24 G cannula, and a stainless steel drip tray (Figure 1). The sterilized EDTA and collagenase buffers were transferred into two 250 mL autoclavable rubber-capped bottles. Prior to starting the perfusion, the buffers were placed in a water bath maintained at 40 °C and oxygenated with 95% O_2_ and 5% CO_2_ for 30 min.

### 3.2. Animals

The study was performed on adult male Wistar rats weighing 200–250 g obtained from the university facility. All animals were housed, fed and treated in accordance with the guidelines of the Institutional Animal Care and Use Committee (IACUC) and with guidelines and regulations of the university. All experimental protocols were approved by the Graduate Studies and Research Committee of the University (IRB Decision No. 91/2023).

### 3.3. Liver Perfusion Process

Rats were anesthetized with intraperitoneal injection of ketamine at a dose of 87 mg/kg body weight combined with xylazine at a dose of 13 mg/kg [2]. The abdominal area was shaved and disinfected with alcohol and iodine. An incision was made and the skin and muscles of the abdomen were carefully opened. After exposing the abdominal organs and identifying the portal vein, a suture was then placed beneath the portal vein, avoiding regions with small blood vessels to minimize the risk of bleeding. The cannula was gently inserted into the portal vein, the needle was removed, and the cannula was secured with the suture in order not to be ejected during perfusion (Appendix A). The successful insertion was confirmed when blood began to flow through the cannula, indicating proper placement (Figure 2A).

The tube was connected to the cannula, and perfusion with EDTA was initiated at a flow rate of 10 mL/min A white spot appeared on the liver, indicating good perfusion. After few seconds, the inferior vena cava (IVC) was cut. Perfusion continued for 10 min until the liver assumed pale color. Gentle pressure was applied to the IVC during the perfusion, causing the liver lobes to enlarge. In the final minute of EDTA buffer perfusion, the flow rate was increased to 22 mL/min.

The tube was immediately switched to the collagenase buffer. During this perfusion, the IVC was temporarily closed for few seconds by clamping with forceps, causing the liver to enlarge; this process was repeated three times. Collagenase perfusion was maintained for 3 min at a flow rate of 22 mL/min after the liver completely assumed a yellowish color (Figure 2B). The pump was then stopped, and the whole liver was removed and placed in a sterilized beaker containing 20 mL of warm collagenase buffer.

### 3.4. Hepatocyte Dissociation and Purification

Using a heating magnetic stirrer plate and a pivot ring stirring bar with a length of 3 cm, gentle stirring (speed 800 rpm) in the warm collagenase buffer opened the liver capsule and facilitated the release and dissociation of hepatocytes from the liver lobes. This process was carried out for 5 min at a moderate speed under a laminar flow hood to maintain sterility. The resulting cell suspension was collected, filtered through a 70–100 µm sterile cell strainer, and centrifuged at 50× *g* for 3 min at 4 °C. The supernatant was discarded, and the pellet was resuspended in 5 mL of washing media and left in ice for 25 min.

Next, 40 mL of cold washing media was added to the remaining liver tissue in the beaker with stirring for 20 min, until all liver lobes were completely digested. The resulting cell suspension was filtered through a 70–100 µm cell strainer, collected in a 50 mL conical tube, and centrifuged at 50× *g* for 3 min at 4 °C using refrigerated benchtop centrifuge. The supernatant was then removed, the pellet was resuspended in 15 mL of washing media and the two resuspended pellets were combined together. The cell suspension was then transferred to a new conical tube containing 20 mL of 90% Percoll or SpermGrad™ in phosphate-buffered saline (PBS, 10×), mixed thoroughly, and centrifuged at 800× *g* for 10 min at 4 °C. The final concentration of Percoll or Spermgrad was 45%. The upper layer was carefully removed, and the pellet (Figure 3A) was resuspended in 20 mL of washing media, followed by centrifugation at 800× *g* for 1 min at 4 °C. Sometimes, the pellet precipitates along the side of the conical tube (Figure 3B). After removing the supernatant, the pellet was resuspended in 20 mL of washing media, centrifuged at 50× *g* for 3 min at 4 °C, and the resulting pellet was resuspended in 15 mL of culture media.

### 3.5. Cell Viability Assay

Cell viability was tested using 0.4% trypan blue stain in 1:1 ratio and measured using a haemocytometer. The viable cells appeared completely yellow while dead cells were blue stained (Figure 4A).

Calculation:

Viable cells count/mL = average number of viable cells per square × 10^4^ × 2 (dilution factor)

Total cells count/mL = counted cells (dead + living) × 10^4^ × 2 (dilution factor)

Viability [%] = $\frac{\mathrm{viable}\;\mathrm{cells}\;\mathrm{count}/\mathrm{mL}}{\mathrm{total}\;\mathrm{cells}\;\mathrm{count}/\mathrm{mL}}$× 100

Note: Sometimes cells may exhibit a yellowish cytoplasm and a blue nucleus (Figure 4B), which can lead to confusion when determining whether they are alive or dead. However, these cells are not healthy for culturing, and culturing them will not yield desirable results.

### 3.6. Hepatocyte Culture and Steatosis Induction

The suspended hepatocytes were seeded in collagen-coated 6-well plates at densities of 7 × 10^5^ and 1 × 10^6^ cells per well, with a volume of 2.5 mL per well, and incubated at 37 °C, 5% CO_2_ for 24 h. in a CO_2_ incubator. After 24 h of culturing, the culture media in plates was removed, the cells were washed twice with 2 mL of 1× PBS, then incubated for 5 h in 2.5 mL per well of starvation media that contained no FBS. Following starvation, the media was replaced with 2.5 mL of lipogenic media (a complex of 0.25 mM BSA and 1 mM sodium oleate) for an additional 24 h for the plates seeded with 1 × 10^6^ cells per well, and for 48 h for the plates seeded with 7 × 10^5^ cells per well [16]. The media was replaced every 24 h and 0.25 mM BSA-containing media was used as a control for the steatotic group because sodium oleate was dissolved in BSA. Two other control groups were used: Unstarved hepatocytes cultured in a standard culture media, and 5 h starved hepatocytes cultured in standard culture media.

### 3.7. Oil Red O Stain

At the end of the experiment, each well was washed twice with 2 mL of 1× PBS by gentle flushing with a dropper, fixed with 2 mL of 4% cold paraformaldehyde for 1 h at room temperature (RT) [17], washed twice with 2 mL of distilled water, then washed with 2 mL of 60% isopropanol for 5 min at RT. The wells were then stained with 1 mL/well of Oil Red O working solution for 15 min at RT, and the plates were covered with aluminum foil to protect from light. The wells were then immediately washed three times with distilled water, stained with 1 mL of hematoxylin for 15 min, rinsed with PBS then kept in PBS for another 15 min before being visualized under the microscope (Micros Austria, MCX1600, Veit/Glan, Austria) [18]. For the quantification test, four random fields from each well were captured at a total magnification of 200× using Leica EC3 digital camera connected to an inverted microscope (Leica Microsystems, Wetzlar, Germany) and Leica application suite LAS X software version 3.7.5.24914 (Leica Microsystems, Zurich, Switzerland). The percentage of the red staining area of each image was analyzed using Image J software version 1.54. For short-term preservation, 2 mL of PBS were added to each well, and the plates were stored at 2–8 °C after paraformaldehyde fixation for up to 24 h before further staining. To preserve the stained cells for a longer time, two drops of melted glycerol jelly were added to each well and covered with coverslips [18].

### 3.8. Statistical Analysis

Data are presented as means ± SEM. Differences were detected using one-way analysis of variance (ANOVA) followed by post hoc Fisher’s least significant difference test (LSD). Differences were considered significant when *p* < 0.05. The experimental data were analyzed using Graph-Pad Prism software version (8.0.1).

## 4. Results

### 4.1. Cell Viability

The total number of harvested cells was ranging between 74,058,330 and 87,000,000 (~75–90 million) cells/rat liver, with cell viability of 86–93% (Figure 5).

### 4.2. Cell Culture and Steatosis Induction

After 18 h of culturing, the cells appeared slightly more cuboidal, assuming a three-dimensional shape and forming aggregates. These clusters of cells often exhibited a cord-like appearance, presumably reminiscent of hepatic trabeculae (Figure 6A). Notably, some of the cultured hepatocytes formed a circular pattern that resembled the cellular architecture surrounding blood vessels (Figure 6A). After 24 h of culturing, the wells seeded with 1 × 10^6^ cells/well reached around 70–80% confluence, while those seeded with 7 × 10^5^ cells/well reached around 50–60% confluence. Figure 6B illustrates the morphology of isolated primary rat hepatocytes after 24 h. By this time, the cells have spread to form a typical monolayer, with well-defined cell edges between adjacent cells. Lipid droplets are present as tiny particles within the cells, and each cell contains one or two centrally located, round nuclei.

### 4.3. Steatotic Hepatocytes

Figure 7 shows small red-stained lipid droplets within the normal hepatocytes in both the unstarved control (Figure 7A) and the 5 h starved control groups (Figure 7B) that were cultured in standard culture media. After 24 h incubation in the lipogenic media, steatosis was effectively induced in the hepatocytes, as evidenced by the accumulation of fat droplets, which were stained in red. Culturing of the 5 h starved control hepatocytes in media containing only 0.25 mM BSA for 24 h (BSA control 24 h group) showed that only small, scattered fat droplets were present within hepatocytes (Figure 7C), whereas the 5 h starved hepatocytes cultured in lipogenic media for 24 h (steatotic 24 h group) showed both large and small fat droplets deposition (Figure 7D). After 48 h of culturing, the same pattern was obtained; 5 h starved control hepatocytes in media containing only 0.25 mM BSA (BSA control 48 h group) exhibited small, scattered fat droplets (Figure 7E), whereas the 5 h starved hepatocytes cultured in lipogenic media (steatotic 48 h group) exhibited an additional increase in fat accumulation (Figure 7F). The fat droplets in these steatotic cells (steatotic 48 h group) became larger and more condensed compared to those observed after 24 h of incubation (Figure 7D).

Fat accumulation in the steatotic hepatocytes (steatotic 24 h and 48 h groups) was significantly higher (*p* < 0.0001) compared to the control hepatocytes (unstarved and the 5 h starved hepatocytes incubated in standard culture media) or the BSA control after 24 h and 48 h of incubation in the lipogenic media (Figure 8). Moreover, fat accumulation in steatotic cells after 48 h was significantly higher (*p* < 0.05) compared to those incubated for 24 h (Figure 8).

## 5. Discussion

The isolation and culturing of primary hepatocytes from rats or mice is a crucial technique for studying liver physiology and pathology. However, it is a challenging process that requires careful optimization of various parameters. Although several published procedures exist, following the same steps does not always guarantee a successful isolation. This may be due to insufficient details in the protocols and/or variations in experimental conditions. A survey conducted by Baker in 2016 [19], involving 1576 researchers, revealed a concerning issue: Over 70% of researchers were unable to replicate another researcher’s work, and more than 50% could not reproduce their own experiments. The lack of reproducibility was attributed to inadequate replication in the laboratory, poor oversight or insufficient statistical power [19]. In this work, we provide a detailed protocol for the successful isolation of viable rat hepatocytes in relatively large numbers that allow faster confluence, thus facilitating physiological and pathological studies. More importantly, this work presents an efficient method for inducing mild and severe steatosis, providing a platform model for studying various pharmacological and molecular approaches to treating or mitigating the disease.

We tested several previously established protocols employing the conventional two-step collagenase perfusion method for primary hepatocyte isolation. Although some of these protocols reported high yields and cell viability, their results were not reproducible, and the number of replicates was often not specified. For example, Shen et al. (2012) [2] reported isolating approximately 1.0 × 10^8^ cells from a single rat liver with viability ranging from 88% to 96%; however, we were unable to reproduce these outcomes in our laboratory using the same procedure. Similarly, Salem et al. (2018) [8] reported isolating 20 × 10^6^ total cells per mouse liver, but this protocol also proved irreproducible in our hands. Other studies have reported comparable yields for mouse liver [9,12], yet they likewise suffer from limited replication. A critical limitation across these reports is the insufficient number of replicate isolations, which undermines the strength of their reports regarding reproducibility. In contrast, our protocol consistently yields 75–90 million viable hepatocytes per rat liver, with cell viability ranging from 86% to 93% in each replicate, thereby demonstrating superior reliability and reproducibility compared to earlier methods.

Several factors affect hepatocyte isolation and culturing process, including pH adjustment, temperature, perfusion route, perfusion flow rate, collagenase concentration, contamination risks, hepatocyte dissociation, viable cell purification, and surface coating of culturing dishes. These variables require fine-tuning to achieve the desired outcomes.

The preparation of buffers and media is a critical step in hepatocyte isolation. Adjusting the pH of the perfusion buffers and lipogenic media to 7.4 before use is essential. Although several protocols [2,12,20] did not mention this detail, simply the use of HEPES buffer does not necessarily guarantee a pH of 7.4. Failing to check and adjust the pH can result in a low cell yield and an unsuccessful cell culture. Therefore, we found that maintaining the buffer sterile, warm, oxygenated, and at a neutral pH is effective in keeping the cells viable and healthy during perfusion. Furthermore, the buffer was supplemented with N-acetyl-L-cysteine, which has been previously reported to have hepatoprotective properties, prevent oxidative damage, and improve cell viability in various cell types [21,22,23,24,25].

The temperature of the perfusion buffers is another critical factor for the success of the experiment, as it directly affects cell viability and collagenase activity. Adjusting the water bath temperature to above 37 °C (typically 40–45 °C) has been recommended in previous studies [2,8,12], as this ensures that the temperature of the outlet buffer at the tip of the cannula remains close to 37 °C. However, this factor can vary depending on lab conditions, such as ambient temperature and the length of the perfusion tubing [12]. Charni-Natan and Goldstein (2020) [9] also suggested that perfusion with warm buffer, which may cool down to 20–25 °C within the tubing, was still sufficient for successful hepatocyte isolation. In our protocol, we set the water bath temperature to 40 °C, with the lab environment maintained at around 25 °C. We also checked the temperature of the buffer after cannulation and before insertion into the portal vein to ensure it was approximately 37 °C, which helped achieving successful cell isolation.

The flow rate of the perfused buffer is another crucial parameter, as both fast and slow flow rates can impact cell viability. High pressure and flow rates can result in hepatocyte death, whereas low pressure and flow rates may lead to insufficient buffer circulation through the smaller blood vessels [26]. Additionally, the duration of perfusion is an important factor; for example, prolonged perfusion may cause hypoxia, while short-duration perfusion may prevent the buffer from achieving the desired effect. Normal and high perfusion rates are defined as 1 and 2 mL/g liver/min, respectively [27]. However, flow rates vary across studies, with some publications not reporting the weight of the animals or the duration of perfusion. These studies often rely on descriptive markers for timing, such as when the liver is “completely cleared of blood” or “when it turns pale and yellow” [2,8,9].

Generally, protocols for mouse liver perfusion, typically for animals weighing 25–30 g or aged 8–10 weeks, set the flow rate at 3–5 mL/min for 4–8 min for both buffers [8,9,11]. For rat weighing approximately 300 g, a flow rate of 10–15 mL/min using EDTA buffer, followed by 25 mL/min for an additional 6 min with collagenase buffer, was recommended [2]. Another protocol for livers of rats weighing 250–300 g specified a perfusion duration of 10–15 min at a flow rate of 15 mL/min for both buffers [7]. A third protocol perfused EGTA buffer at 40 mL/min for 4 min and collagenase buffer at 20 mL/min for 6.5 min in rats of the same weight range [11]. In the current protocol, we found that a flow rate of 10 mL/min for 10 min with EDTA buffer was sufficient. During the last minute of perfusion with EDTA, the flow rate was increased to 22 mL/min and switched with the collagenase buffer for an additional 3 min. This procedure was effective for rats weighing 200–250 g. These settings effectively cleared the liver of blood and loosened the junctions between cells, enabling efficient cell dispersion.

Several previous protocols used the retrograde perfusion through IVC to isolate hepatocytes [8,9,12]. However, in this study, we favored anterograde perfusion through the portal vein, as it provides hepatocytes with enhanced proliferative capacity [27,28] and superior quality for culture. In support, Yin et al., 2007 [29] noted that there was a difference in regeneration and proliferation capacities between cells from periportal and pericentral region in the liver. They used this observation to support their finding that anterograde method led to a higher engraftment capacity compared to the retrograde method. Anterograde perfusion was thought to yield a higher proportion of hepatocytes from zone I, the periportal region, because it was more oxygenated and provided cells with a higher proliferative capacity.

Successful cannulation is an important step in an in vivo perfusion process. In this protocol, we indicated that the successful cannulation can be confirmed by four successive indicators: First, initial confirmation occurs when blood begins to flow through the cannula upon proper insertion into the portal vein. Second, after attaching the tube to the cannula and starting the flow of perfusion, a white spot appears on the surface of liver lobes, indicating that the buffer has entered the liver lobe [9]. Third, following IVC cutting, liver lobes inflate when the IVC is temporarily clamped [2,9]. Fourth, the absence of tissue swelling around the portal vein suggests that the perfusion buffer is taking the normal bath as in the circulatory system.

Selection of the type of collagenase is very critical which depends on the type of investigation to be conducted on the isolated hepatocytes. In studies where a large number of cells is required to evaluate intracellular proteins, collagenase type I is the preferred enzyme. Conversely, when maintenance of the integrity of surface and transmembrane proteins along with their downstream signaling molecules is essential, gentler collagenases, such as collagenase IV, are more appropriate [13]. Furthermore, the selection of the collagenase concentration is essential. For crude collagenase, the optimal concentration was found to be 300 µg/mL, whereas liberase, a mixture of collagenase I and II, was effective for hepatocyte isolation at a final concentration of 25–40 µg/mL [9,30]. Type IV collagenase was found to be effective for hepatocyte isolation at a final concentration of 0.05% (wt/v), which is equivalent to 500 µg/mL [10]. In this study, we found that collagenase IV at concentrations of 100, 400, or 500 µg/mL effectively facilitated cell dispersion. However, the optimal concentration for hepatocyte isolation in our protocol was 400 µg/mL.

Maintaining sterile conditions is crucial at every stage of perfusion and culturing. Contamination remains one of the major challenges in primary cell culture [5,12]. In our protocol, we ensured that all laboratory equipment and tools were either autoclaved or provided as sterile disposable materials. Additionally, we found that using a sterile intravenous set instead of autoclavable tubing [8,26], along with incorporating 1% antibiotic-antimycotic mixture in perfusion buffers and media [8,12] was highly effective in preventing contamination during hepatocyte isolation and culture.

The release of hepatocytes from the perfused liver using mechanical stress methods such as stretching and tearing the tissue with forceps, pitting the tissue, cutting it into pieces, shaking, or using a cell scraper to disperse the cells has been documented [2,8,9,12]. However, we found that these methods were inefficient in obtaining a high number of viable cells. In our protocol, hepatocytes were dissociated gently and gradually; first, the connective tissue of the liver capsule was disintegrated using warm collagenase with stirring using a magnetic stirrer bar under laminar flow. The hepatocyte release was then completed by continuing to stir in a washing buffer. This method consistently produced a high yield of viable cells. A high yield of cells allows researchers to test numerous parameters, perform multiple replicates per test, and preserve cells through cryopreservation for future use.

The liver is composed of approximately 80% hepatocytes, with the remaining cells classified as non-parenchymal cells, including liver sinusoidal endothelial cells, stellate cells, and Kupffer cells [31]. Purification of parenchymal cells, specifically hepatocytes, from non-parenchymal cells, and the separation of viable hepatocytes from dead ones, is a crucial step in liver cell research. The most recent successful protocols for isolating hepatocytes yielded 18–50 × 10^6^ cells per mouse liver (8–14 weeks old), with a viability rate of ~ 93–96% [8,9,12,14], and up to 100 × 10^6^ cells per rat liver preparation, with viability rates ranging from 88% to 96% [2]. Berardo et al. (2020) [13] isolated around 150 to 200 × 10^6^ cells per rat liver preparation, with viability rate of 80%. However, it was estimated that a mouse liver contains approximately 135 ± 10 million hepatocytes per gram of liver tissue, while a rat liver contains 117 ± 30 million hepatocytes per gram of liver [32]. This indicates that these protocols isolated less than 50% of the total hepatocytes in the liver, underscoring the importance of the purification process. A density gradient solution, such as Percoll or spermGrad, is recommended to remove dead hepatocytes and other contaminating cells. Failure to use a purification step prior to cell culturing can result in unsuccessful cultures, as dead cells are heavy and occupy space, preventing viable hepatocytes from adhering to the surface to form a monolayer. In our protocol, we successfully isolated 74–87 million hepatocytes from rat liver, with a cell viability of 86–93%. The isolated hepatocytes were healthy and functional in culture for more than 72 h.

It is important to highlight that ex vivo perfusion of rodent livers using sophisticated and semi-automated devices, such as the gentle MACS Perfuser [5] or integrated perfusion systems [26], which controlled multiple perfusion parameters, significantly improved hepatocyte yield compared to the conventional method. For instance, Ng et al. (2021) [26] successfully isolated 200–500 million cells per rat liver (rats weighing 200–300 g), with a high cell viability rate of 85–95%, without requiring density gradient separation. Similarly, Poggel et al. (2022) [5] reported isolating 15–59 million cells per gram of liver, with cell viability ranging from 81% to 94%. While these sophisticated devices offer greater optimization, control over multiple parameters, consistency of results, the availability of these instruments, their cost and maintenance can pose challenges to many laboratories. Nevertheless, the conventional perfusion method, such as the one we used, remains cost-effective, can be optimized, and is capable of yielding a sufficient number of cells, especially for experiments involving freshly isolated cells.

Notably, this protocol can be used to isolate primary hepatocytes from mice with some technical modifications. The technical differences in isolating primary hepatocytes from mice compared to those of rats are primarily related to parameters pertinent to the size of the animal and its liver tissue and vessels. These adjustments include using a smaller needle gauge (26 G), thinner perfusion tubing, and reducing the flow rate to 3 mL/min [9]. Additionally, in mice, it is more convenient to perform the perfusion via the inferior vena cava, as it is larger than the portal vein [9,12]. However, no significant differences have been reported in the functional activity of hepatocytes isolated from mice compared to those from rats.

Some researchers also stressed the necessity of coating of culture dishes and wells with collagen or other materials to allow hepatocyte growth and proliferation while others showed that it is unnecessary [2,12]. We showed that hepatocytes successfully adhered to and proliferated on various surfaces, including collagen-coated plates and coverslips, uncoated Petri dishes with untreated surfaces, and uncoated 6-well plates treated with Nunclon™ Delta surface treatment. However, collagen coating significantly enhanced hepatocyte attachment, particularly in experiments involving multiple staining and rinsing steps with different solutions and in calcium imaging measurement technique.

For successful culturing of hepatocytes, there are some tricky steps that should be carefully taken into consideration during seeding. For instance, the number of cells cultured should be based on the number of viable cells only, not the total number of cells. Furthermore, to ensure that each well contains approximately similar number of cells, gently swirl the conical tube to make a homogeneous suspension of cells before pipetting and seeding each well, as hepatocytes are heavy and precipitate quickly. Researchers recommended culturing cells at a density of 4 × 10^5^–5 × 10^5^ cells per well in a 6-well plate when cell viability is above 85% [2,9,12]. This allowed the cell density to reach approximately 60–70% confluence promoting cell–cell contact while maintaining sufficient space for the hepatocytes to grow to their full size, ultimately achieving 90–95% confluence after 24 h [2]. However, we found that seeding with 7 × 10^5^ to 1 × 10^6^ cells per well in 6-well plates resulted in higher confluence and formation of a monolayer, which was good for short-term experiments (e.g., 12–24 h) to avoid cell dedifferentiation [33,34]. The isolated primary rat hepatocytes, in this study, were characterized by their distinct morphology in culture. During the early stages of culturing, hepatocytes displayed normal polar morphology; they had a 3D structure with an apical surface and did not exhibit a flattened shape. They had a slightly cuboidal shape, aggregating into clusters and forming cord-like structures which could be reminiscent of hepatic trabeculae (Figure 6A). Our observations are aligned with the findings of a previous study which described the morphological characteristics of differentiated rat hepatocytes in vitro [1]. Interestingly, some of our cultured hepatocytes formed a circular arrangement that exhibited a cellular architecture similar to that of hepatocytes surrounding the blood vessels in liver tissue. After 24 h of culturing, the hepatocytes in the current study exhibited polygonal morphology, with cells closely contacting each other to form a highly confluent monolayer. Interestingly, we observed a high number of binucleated hepatocytes in the culture. This phenomenon, known as somatic polyploidization, is a physiological process in hepatocytes occurring at specific stages of development and believed to be essential for increasing metabolic output, cell and organ size, and maintaining specialized cellular functions [35]. Approximately 20–30% of hepatocytes were binucleated, presumably a result of failed cytokinesis [36].

It is important to assess the quality and functional activity of cultured hepatocytes. Actually, the functional activity of the isolated hepatocytes is assessed by the following observations: when seeded at an appropriate density in a suitable environment, hepatocytes are able to adhere to the surface, form physical contacts, and establish approximately 50% confluence within 24 h. They are also capable of dividing, growing to fill the available space, and responding to external stimuli, such as nutrient uptake, which is demonstrated in the current study by intracellular lipid accumulation. Conversely, a loss of the ability to adhere and to form approximately 50% confluence within 24 h after seeding indicates that the cells are non-viable and functionally impaired.

In this study, steatosis was induced in cultured rat hepatocytes. Steatosis is commonly induced in vitro using various methods, such as incubating cells in media containing a combination of sodium palmitate and sodium oleate [17,20], or by pre-incubating cells in starvation media followed by incubation in high-glucose media with 30 mM glucose and 100 nM insulin [18]. Furthermore, cyclosporine A was used as a steatotic inducer [37]. Although these methods are effective in inducing fatty liver disease in cultured hepatocytes, some hepatotoxicity was associated with the inducing agents such as palmitate and cyclosporine A [20,38]. In our protocol, steatosis was induced efficiently by pre-starving hepatocytes for 5 h, followed by incubation in lipogenic media containing 1 mM sodium oleate and 0.25 mM BSA. Pre-starvation for 5 h effectively seemed to enhance lipid uptake [16,18]. Neutral lipids were specifically stained with Oil Red O for quantification analyses, which showed a time-dependent increase in lipid deposition (Figure 7). After 24 h of incubation in lipogenic media, hepatocytes exhibited a 21% increase in lipid accumulation (mild steatosis) compared to the 3 control groups (Figure 8). Lipid accumulation increased further to produce severe steatosis in hepatocytes incubated for 48 h (Figure 7F), compared to those incubated for 24 h (Figure 7D). Previous studies showed that incubation of primary rat hepatocytes in 1 mM sodium oleate effectively stimulated low-grade steatosis by increasing triglyceride levels with low cytotoxicity, and comparable results were obtained using a 1:1 mixture of sodium oleate and sodium palmitate after 24 h of incubation [20]. Lin et al. (1995) [16] demonstrated that pre-starvation of human and rat primary hepatocytes, as well as HepG2 cells, for 2 h, followed by incubation in media containing 0.25 mM BSA and 1 mM oleate for 5 h, effectively increased cellular triglyceride accumulation.

Collectively, this communication provided a detailed description of critical steps in the isolation and culture of hepatocytes, including the adjustment of pH of buffers, media, and solutions prior to use, identifying markers for proper cannulation and perfusion, and recognizing features of viable cells suitable for culturing. We enhanced the technique by employing a gentle, gradual method to release hepatocytes from liver tissue, rather than the mechanical stress previously described [9]. Additionally, we discussed best practices for culturing and handling hepatocytes. Furthermore, this work presented an effective protocol for inducing mild and severe steatosis, along with improvements in lipid staining quality to facilitate the detection of lipid accumulation in hepatocytes.

## 6. Conclusions

Primary hepatocytes are an indispensable in vitro tool to study the physiology and pathology of liver, and the isolation of viable and functional hepatocytes is highly demanded. The protocols for in situ perfusion and isolation are regularly improved by adjusting and standardizing many parameters to increase the success rate of isolation and culturing processes as well as enhancing the reproducibility of the procedure. In this work, we provide guidelines for the successful harvesting and culturing of large number of functional hepatocyte as well as inducing steatosis in these cells as a prerequisite for the study of lipid metabolism.

## 7. Reagents Setup

### Solutions, Media and Culture Plates

EDTA buffer: This buffer was prepared by dissolving the following components: 0.5 mM EDTA, 10 mM HEPES, 5 mM N-acetyl- L-cysteine, and 10 mM D-glucose in HBSS without Ca^2+^, Mg^2+^, and phenol red (1×), and 1% antibiotic-antimycotic solution.

Collagenase buffer: This buffer was prepared by dissolving 10 mM HEPES, 5 mM N-acetyl- L-cysteine in HBSS with Ca^2+^ and Mg^2+^, without phenol red (1×), and 1% antibiotic-antimycotic solution. 0.04% collagenase IV was added later during the perfusion of EDTA buffer.

Both EDTA and collagenase buffers were prepared fresh, and pH was adjusted to 7.4 and filtered using a 0.22 µm sterile syringe filter.

Washing media: 1× DMEM low glucose containing 5% FBS and 1% antibiotic-antimycotic solution.

Culture media: William’s medium E (1×) + GlutaMAX™ containing 6% FBS, 1% antibiotic-antimycotic, and 10 µg/mL insulin.

Starvation media: William’s medium E (1×) + GlutaMAX™ containing 1% antibiotic-antimycotic, and 10 µg/mL insulin.

Lipogenic media: William’s medium E (1×) + GlutaMAX™ containing 0.25 mM BSA (fraction V, fatty acid free, or bovine serum albumin lyophilized containing ~1% fatty acid), 1 mM sodium oleate, 1% antibiotic-antimycotic, and 10 µg/mL insulin. Lipogenic media was freshly prepared and the pH was adjusted to 7.4 and filtered using a 0.22 µm sterile syringe filter [16].

Fixative solution (4% paraformaldehyde): This solution was prepared by dissolving 4 gm of paraformaldehyde in 80 mL of 1× PBS using a magnetic stirrer plate at 60 °C under a ventilation hood. 1 M of sodium hydroxide was added dropwise until the solution became clear. After the paraformaldehyde was dissolved and cooled, the pH was adjusted to 6.9 using 1% HCl. The volume of the solution was then adjusted to 100 mL with 1× PBS. The solution was filtered, aliquoted and frozen or stored at 2–8 °C.

Oil Red O Stain (stock solution 0.5%): 0.5 g of Oil Red O stain was dissolved in 100 mL of isopropanol and heated to 60 °C to enhance the solubility. The stock solution was stored at room temperature (25 °C) in a dark bottle away from direct light. Working solution was freshly prepared by diluting the stock solution with distilled water at a 3:2 ratios (3 mL of stock solution and 2 mL of distilled water). The working solution was filtered and allowed to stand for at least 10 min at room temperature [18].

Culture Plates

6-well-plates were used and the wells were coated with collagen Type I (1:10 in PBS) at a concentration of 10 µg/cm^2^. The coating was allowed to dry for two h, then kept at 4 °C overnight. On the day of the experiment, the plates were sterilized by exposure to UV light for 1–3 h, then washed with sterile culturing media before use.

## Figures and Tables

**Figure 1 mps-08-00111-f001:**
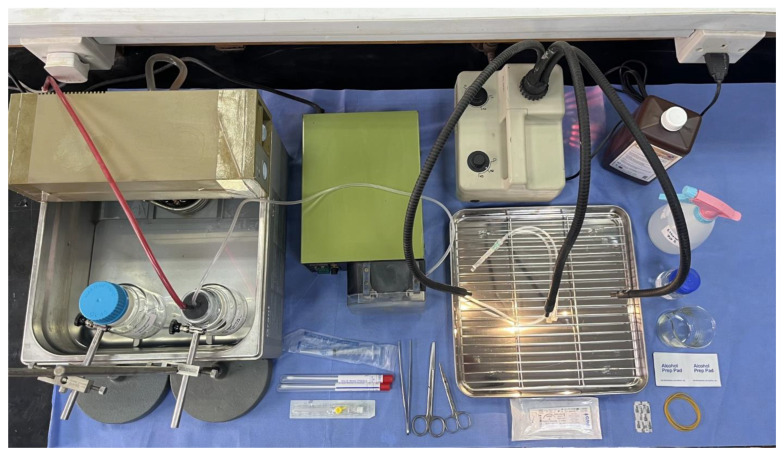
The perfusion setup. All the materials and tools were sterilized and kept in a clean, airflow-free environment.

**Figure 2 mps-08-00111-f002:**
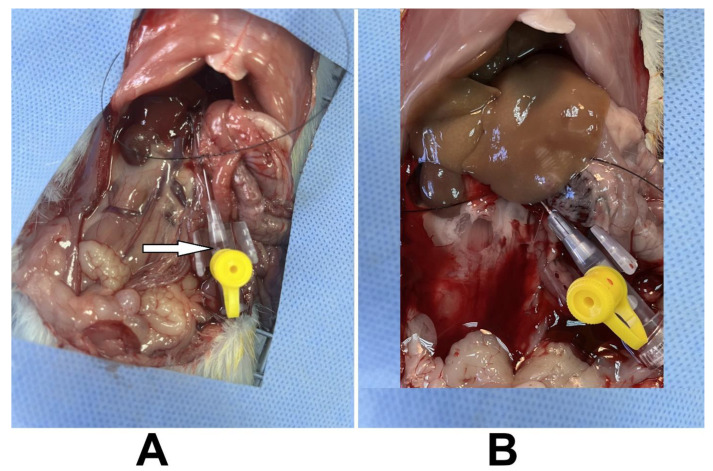
Successful cannulation. (**A**) Insertion of the cannula into the portal vein, a first step of the liver perfusion process. The arrow shows the blood flow through the cannula, confirming correct placement. (**B**) Liver after perfusion with EDTA and collagenase buffers. The liver assuming a pale color indicating a successful perfusion.

**Figure 3 mps-08-00111-f003:**
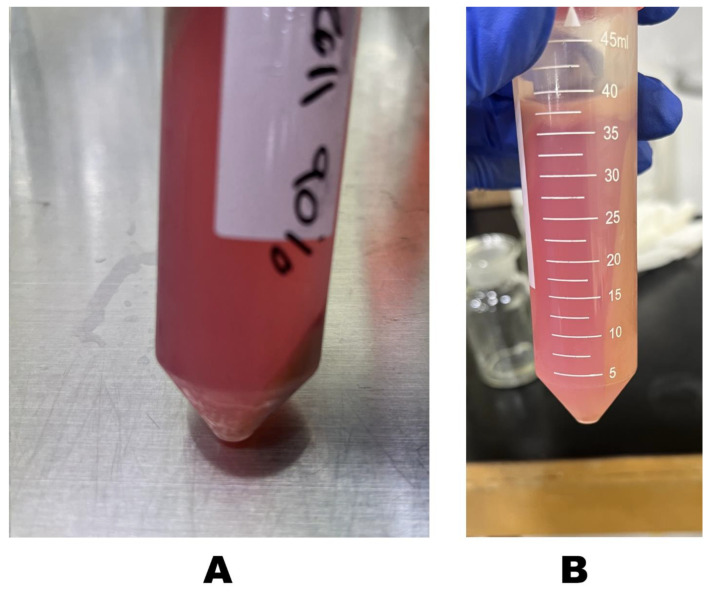
Purification of viable hepatocytes using Percoll as a density-gradient solution. The pellet after centrifugation precipitates at the bottom of the conical tube (**A**), but sometimes the pellet precipitates along the side of the conical tube (**B**) and in this case, it needs re-centrifugation to remove as much as possible of dead cells.

**Figure 4 mps-08-00111-f004:**
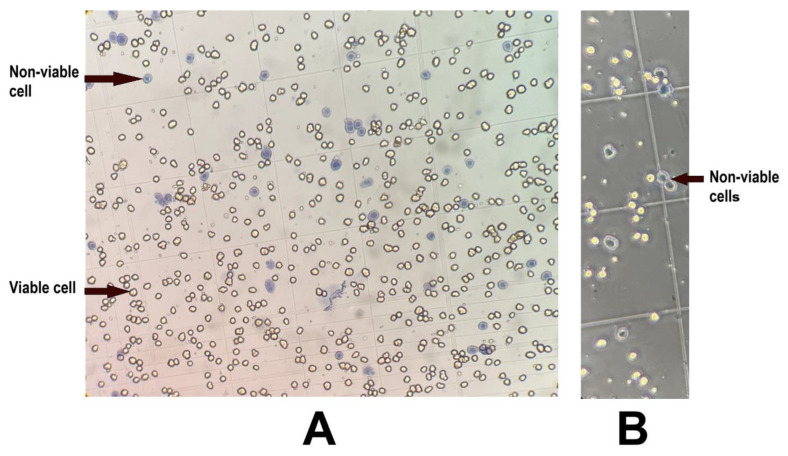
Cell viability assessment using trypan blue stain. (**A**) Viable cells appeared completely yellow and assumed a rounded shape, while dead cells stained blue and assumed a flattened appearance. (**B**) Cells with a blue-stained nucleus and somewhat yellow cytoplasm (arrow) were considered non-viable and unsuitable for culturing.

**Figure 5 mps-08-00111-f005:**
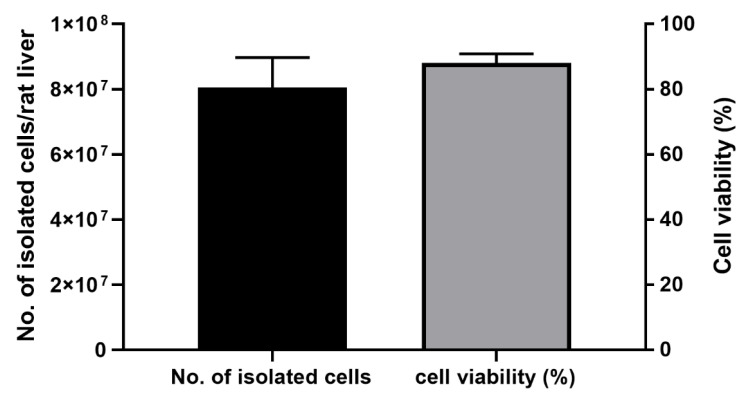
The number and viability of isolated primary rat hepatocytes. Cells were harvested using 45% Percoll density gradient centrifugation. Viable cells were detected after staining with 0.4% trypan blue and enumeration using a haemocytometer. The number of independent primary rat hepatocyte isolations from whole liver samples was three.

**Figure 6 mps-08-00111-f006:**
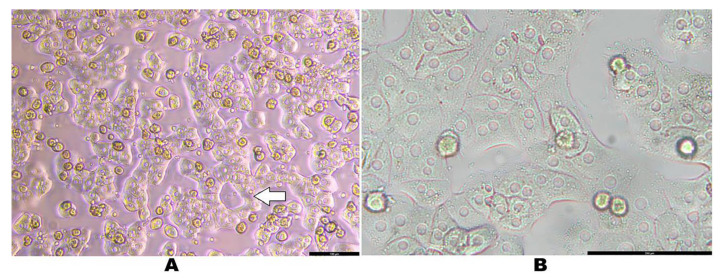
Normal morphology of isolated primary rat hepatocytes. (**A**) A phase contrast image of cultured hepatocytes after 18 h at a seeding density of 7 × 10^5^ cells/well, with confluence greater than 50%. Scale bar = 100 µm. Total magnification = 100×. (**B**) A bright field image of cultured hepatocytes after 24 h at a seeding density of 1 × 10^6^ cells/well, with confluence greater than 80%. Scale bar = 100 µm. Total magnification = 200×. The white arrow indicates a cluster of cells forming a circular pattern.

**Figure 7 mps-08-00111-f007:**
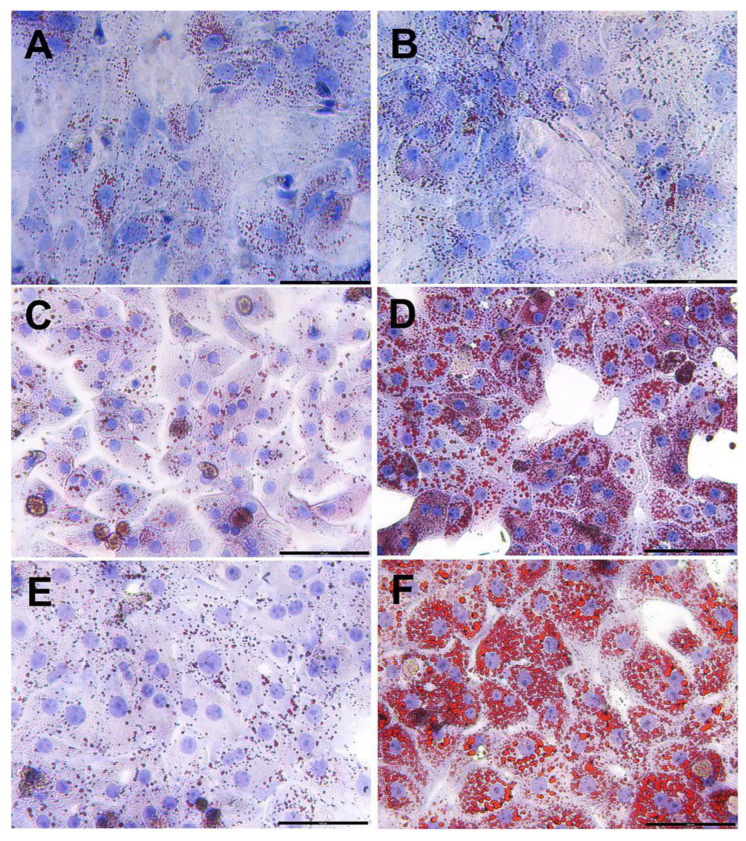
Induction of steatosis in isolated primary rat hepatocytes. (**A**) Unstarved control hepatocytes cultured in standard culture media, (**B**) 5 h starved control hepatocytes cultured in standard culture media, (**C**) BSA control 24 h, 5 h starved hepatocytes cultured in media containing only 0.25 mM BSA for 24 h. (**D**) Steatotic 24 h, 5 h starved hepatocytes cultured in lipogenic media containing 1 mM sodium oleate-0.25 mM BSA for 24 h, (**E**) BSA control 48 h, 5 h starved hepatocytes cultured in media containing only 0.25 mM BSA for 48 h, (**F**) Steatotic 48 h, 5 h starved hepatocytes cultured in lipogenic media containing 1 mM sodium oleate-0.25 mM BSA for 48 h. Total magnification = 200×, Scale bar = 100 µm.

**Figure 8 mps-08-00111-f008:**
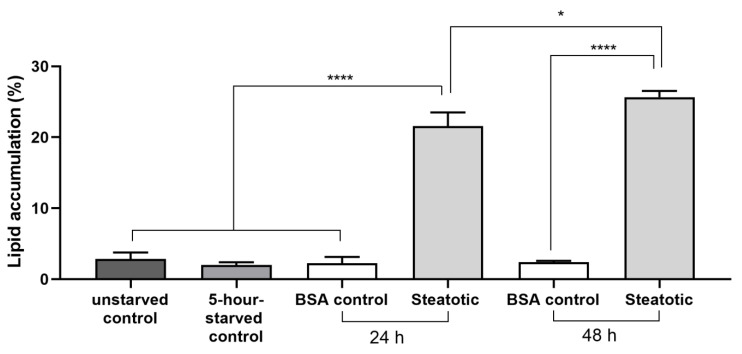
Percentage of lipid accumulation in primary rat hepatocytes after incubation in lipogenic media for 24 and 48 h. Four random fields of each experiment were captured and analyzed using image J software. Steatotic hepatocytes are those that accumulated lipid droplets after incubation in lipogenic media. Unstarved control group (N = average of 8 fields), 5 h starved control group (N = average of 8 fields), BSA control 24 h (N = average of 8 fields), steatotic 24 h (N = average of 16 fields), BSA control 48 h (N = average of 20 fields), steatotic 48 h (N = average of 52 fields). Data are expressed as means ± S.E.M. and were analyzed by one-way ANOVA followed by Fisher’s LSD test. **** *p* < 0.0001 indicates a significant difference compared to the relevant controls: unstarved, 5 h starved and BSA control groups, * *p* < 0.05 indicates a significant difference between steatotic cells cultured in lipogenic media for 24 compared to 48 h.

## Data Availability

All data generated during this work are embedded in this manuscript. The corresponding author will provide any further details upon request.

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
