# Peer review of "Optimized Protocol for Primary Rat Hepatocyte Isolation and a Model for Investigating Experimental Steatosis"

_mps, 2025, doi:10.3390/mps8050111_

Round 1
Reviewer 1 Report
Comments and Suggestions for Authors
The manuscript presents a detailed protocol for isolating primary rat hepatocytes and establishing an in vitro steatosis model. The study presents a technically challenging area and provides valuable troubleshooting guidance. However, some problems should be addressed.
Major comments:
- The protocol largely follows established two-step collagenase perfusion. Although the discussion justifies choices in theory but lacks data to support this protocol outperforms established methods, including pH, temperature, perfusion flow rate, collagenase concentration, etc. (line 372~374). Please, provide comparative data showing superiority over a "standard" or commonly used protocol in yield and viability.
- The authors stated that N-acetyl-L-cysteine (NAC) was added in buffer, and claimed that it had "hepatoprotective properties and prevents oxidative damage" to maintain cell viability (line 381-383). However, no experimental evidence is provided to demonstrate that NAC actually improved viability or reduced oxidative stress.
- Statements like " reproducibility " (line 578) and yield ranges (line 468, line 28-29, et.al.) remain anecdotal without n-values. Please proved the number of rats and independent isolations for yield and viability data.
- Is it sufficient to reliance solely on Oil Red O (ORO) area (%) to define the severity of steatosis? Claims of "mild" vs "severe" steatosis are unsubstantiated (line 369).
Minor comments:
- Abstract: (line 33-35) “the protocol offers an effective in vitro model for studying the pathophysiology of fatty liver disease” is overstated without functional validation data.
- Line 46-50: The link between isolation optimization and studying specific receptors (STIM1, TRPV1) are mentioned in Introduction, but not revisited in the results related to the steatosis model. The study did not conduct related work, so such a detailed enumeration is unnecessary.
Author Response
Responses to comments from Reviewer 1
Responses to major comments:
Comment 1:
The protocol largely follows established two-step collagenase perfusion. Although the discussion justifies choices in theory but lacks data to support this protocol outperforms established methods, including pH, temperature, perfusion flow rate, collagenase concentration, etc. (line 372~374). Please, provide comparative data showing superiority over a "standard" or commonly used protocol in yield and viability.
Response:
For each parameter, we have already discussed the variability present in the established protocols, which rendered these protocols less successful in achieving high yield and high cell viability. We also described how we fine-tuned each parameter to enhance reproducibility.
Additionally, we have now included the following paragraph in the manuscript (lines 380-390, pages 12-13) to summarize these adjustments:
“Although this protocol follows a conventional isolation method, we have made precise adjustments to several parameters, allowing for improved reproducibility and higher yields of viable cells compared to previous protocols. Specifically, we adjusted the buffer pH to 7.4, maintained strict temperature control (40 °C water bath and 25 °C ambient temperature), performed anterograde perfusion via the portal vein, and set the flow rate to 10 mL/min for 10 minutes for the EDTA buffer and 22 mL/min for 3 minutes for the collagenase buffer. Hepatocyte dissociation was achieved through gentle stirring in warm collagenase buffer. A 45% density gradient solution (either ‘Percoll’ or ‘SpermGrad’) was used to purify viable cells. To prevent contamination, antibiotic and antimycotic solutions were added to the buffers. Collagen coating was applied to improve hepatocyte adhesion during culture.”
Comment 2:
The authors stated that N-acetyl-L-cysteine (NAC) was added in buffer, and claimed that it had "hepatoprotective properties and prevents oxidative damage" to maintain cell viability (line 381-383). However, no experimental evidence is provided to demonstrate that NAC actually improved viability or reduced oxidative stress.
Response:
N-acetylcysteine (NAC) is a well-known antioxidant that reduces oxidative stress. Numerous studies have demonstrated its effectiveness. There are several experimental articles showing that NAC reduces oxidative stress and improves cell viability across various cell types (Sagristá et al., 2002; Yedjou and Tchounwou, 2007; Partyka et al., 2015; Hosseini et al., 2019). We assume that we do not have to prove that in our experiments or to repeat what the literature cited. We just benefited from published work.
- Sagristá ML, GarcÍa AF, De Madariaga MA, Mora M. Antioxidant and pro-oxidant effect of the thiolic compounds N-acetyl-L-cysteine and glutathione against free radical-induced lipid peroxidation. Free Radical Research. 2002 Jan 1;36(3):329-40.
- Yedjou CG, Tchounwou PB. N-acetyl-l-cysteine affords protection against lead-induced cytotoxicity and oxidative stress in human liver carcinoma (HepG2) cells. International Journal of Environmental Research and Public Health. 2007 Jun;4(2):132-7.
- Partyka A, Niżański W, Bratkowska M, Maślikowski P. Effects of N-acetyl-L-cysteine and catalase on the viability and motility of chicken sperm during liquid storage. Reproductive Biology. 2015 Jun 1;15(2):126-9.
- Hosseini E, Ghasemzadeh M, Atashibarg M, Haghshenas M. ROS scavenger, N‐acetyl‐l‐cysteine and NOX specific inhibitor, VAS2870 reduce platelets apoptosis while enhancing their viability during storage. Transfusion. 2019 Apr;59(4):1333-43.
Comment 3:
Statements like " reproducibility" (line 578) and yield ranges (line 468, line 28-29, et.al.) remain anecdotal without n-values. Please proved the number of rats and independent isolations for yield and viability data.
Response:
The sentence “The number of independent primary rat hepatocyte isolations from whole liver samples was three (N=3)” has been added and highlighted in lines 297-298, page 9.
Comment 4:
Is it sufficient to reliance solely on Oil Red O (ORO) area (%) to define the severity of steatosis? Claims of "mild" vs "severe" steatosis are unsubstantiated (line 369).
Response:
First, while staining procedures alone are not sufficient to accurately define the severity of steatosis, histological analysis remains a powerful tool for assessing this condition in histopathology. A clinical scoring system is commonly used to evaluate the degree of steatosis. In this protocol, the severity (mild or severe) was based on the percentage and size of intracellular lipid droplets accumulated within hepatocytes. Using ImageJ software for analysis of lipid accumulation, we considered a low percentage and small lipid droplet size as mild steatosis, while a high percentage and large lipid droplet size as severe steatosis.
Responses to Minor comments:
Comment 1:
Abstract: (line 33-35) “the protocol offers an effective in vitro model for studying the pathophysiology of fatty liver disease” is overstated without functional validation data.
Response:
The lipid staining data remain valid, and the lipid was clearly visible to the naked eye as a white layer and turbidity in the cell lysate samples of steatotic group. Furthermore, cholesterol and triglyceride levels were measured using commercial colorimetric assay kits. The results showed higher levels of both cholesterol and triglyceride in the steatotic group compared to the control group in cell lysates. The data from such analysis are to be reported in an ongoing study, as well as data on TRPV1.
Comment 2:
Line 46-50: The link between isolation optimization and studying specific receptors (STIM1, TRPV1) are mentioned in Introduction, but not revisited in the results related to the steatosis model. The study did not conduct related work, so such a detailed enumeration is unnecessary.
Response:
True, we mentioned the connection between the optimization of the isolation protocol and the study of specific receptors, but the experimental model of primary hepatocytes and steatosis developed in the current protocol is now being used to investigate TRPV1 expression and functionality.
Reviewer 2 Report
Comments and Suggestions for Authors The manuscript of Harb et al. describes an effective protocol for the successful isolation, culturing and lipid loading of rat primary hepatocytes. This manuscript was a pleasure to read, with only very minor grammatical issues. The introduction described the confusion and inconsistencies of published protocols, perhaps leading to poorer outcomes. I appreciated that the issues raised described a number of the common problems. The results clearly showed successful isolations of high purity and viability. Best of all, I appreciated the discussion, which took each step of the protocol and compared published methods with the proposed method in the manuscript. Overall, this is an excellent manuscript. I can find no faults or omissions in the protocol and I expect this protocol will become a gold standard protocol in the field. Congratulations to the authors. I do have a few small comments for the authors to consider.- Could the authors please comment on any differences (technical or functional) between the protocol in rat and mice.
- One of the biggest technical challenges for researchers new to this protocol, is the insertion of the needle into the portal vein. It would be helpful if the authors could include more (or different, or closer up) images (Figure 2) with the portal vein more clearly indicated, organ landmarks, direction of insertion.
- It would be helpful if the authors could include a standard functional assay on the cultured hepatocytes as an appropriate measure of the quality of the cells and a proof that the cells are functionally active.
Author Response
Responses to comments from Reviewer 2
Thank you for your positive and encouraging feedback. we sincerely appreciate your time, effort, and thoughtful comments.
Comment 1:
Could the authors please comment on any differences (technical or functional) between the protocol in rat and mice.
Response:
Sure, we have added the following paragraph in lines 518-526, page 15:
“Notably, this protocol can be used to isolate primary hepatocytes from mice with some technical modifications. The technical differences in isolating primary hepatocytes from mice versus rats are primarily related to parameters pertinent to the size of the animal and its liver tissue and vessels. These adjustments include using a smaller needle gauge (26 G), thinner perfusion tubing, and reducing the flow rate to 3 mL/min [12]. Additionally, in mice, it is more convenient to perform the perfusion via the inferior vena cava, as it is larger than the portal vein [12, 15]. However, no significant differences have been reported in the functional activity of hepatocytes isolated from mice compared to those from rats.”
Comment 2:
One of the biggest technical challenges for researchers new to this protocol, is the insertion of the needle into the portal vein. It would be helpful if the authors could include more (or different, or closer up) images (Figure 2) with the portal vein more clearly indicated, organ landmarks, direction of insertion.
Response:
We have provided a supplementary figure (S1) with its legend illustrating the step-by-step process of needle insertion into the portal vein (Line 187, page 5).
Supplementary Figure 1: Steps for successful insertion of the cannula into the portal vein
Step 1: After opening the abdomen and gently pushing the visceral organs to the right, the portal vein becomes visible and is loosely surrounded by a nylon suture loop.
Step 2: The tissue surrounding the portal vein is gently stretched downward using the index and middle fingers of the researcher’s left hand to keep the vein taut and straight. The researcher’s right hand holds the cannula over the two fingers of the left hand to stabilize it and prevent shaking during insertion. The cannula is aligned with the direction of the portal vein.
Step 3: The tip of the cannula is gently inserted into the portal vein while simultaneously withdrawing the needle from the catheter (plastic tube).
Step 4: Once the needle is carefully removed, blood begins to flow through the catheter.
Step 5: The catheter is carefully advanced further into the portal vein and securely tied with the suture.
Step 6: A close-up view showing successful catheter insertion, evidenced by blood flow through the catheter.
Comment 3:
It would be helpful if the authors could include a standard functional assay on the cultured hepatocytes as an appropriate measure of the quality of the cells and a proof that the cells are functionally active.
Response:
We have added the following sentences in the manuscript (lines 565-573, page 16)
“It is important to assess the quality and functional activity of cultured hepatocytes. Actually, the functional activity of the isolated hepatocytes is evidenced by the following observations: when seeded at an appropriate density in a suitable environment, hepatocytes are able to adhere to the surface, form physical contacts, and establish approximately 50% confluence within 24 hrs. They are also capable of dividing, growing to fill the available space, and responding to external stimuli, such as nutrient uptake, which is demonstrated in the current study by intracellular lipid accumulation. Conversely, a loss of the ability to adhere and to form approximately 50% confluence within 24 hours after seeding indicates that the cells are non-viable and functionally impaired.”

Reviewer 3 Report
Comments and Suggestions for Authors
In this paper by Amani A. Harb et al., the authors describe in detail a protocol for the isolation of rat hepatocytes by the in situ liver perfusion method. In their description, they indicate which steps to follow and the critical points to take into consideration to optimize this technique and successfully isolate and maintain rat hepatocytes in culture, reducing technical errors as much as possible. Furthermore, they describe their in vitro model of mild and severe steatosis for some studies of lipid metabolism.
This article is well written and describes in detail the steps to follow, so as to give the possibility to those who already have experience in this type of procedure to improve their technique, but it also offers a step-by-step guide for those who are less experienced or are approaching this technique for the first time.
However, there are some points that the authors should specify better:
- COLLAGENASE: they use 0,04% (wt/v) or 400 µg/ml. How many units does it correspond to? Is it always the same or have they noticed differences between the various batches? Have you tried other brands besides the one indicated?
- In Methods section, culture plates paragraph: It is useful they specify the total amount of collagen per cm2, thus allowing to repeat the process also for smaller or larger surface areas. They specify the collagen Type I and they write “(1:10)”. Does this mean they diluted the collagen 1:10? In what, PBS? The product they indicate is concentrated 6 mg/ml according to the Sigma-Aldrich website. If they dilute it 1:10, do they use it 0.6mg/ml? It is just to understan whether they use approximately 170 µl/well in a 6 well plate or they perform an intermediate dilution.
- In the Hepatocyte Dissociation and Purification section, they use a magnetic stirrer for digested liver dissociation. Does the size and the shape of the stirrer bar influence the outcome (e.g. cylindrical or pivot ring stirring bar)? What is the meaning of “gentle stirring”? The minimum possible?
- What is the diameter and the lenght of tubing they use? Does the tubing lenght and diameter influence the outcome of the perfusion?
- Can this procedure be used to isolate murine hepatocytes? Obviously the flow rate need adjustments, but did the authors use this procedure for mouse livers?
- In Discussion, lines 366-367: please specify that are rat hepatocytes
- In figure 7 A and B: The hepatocytes have a slightly different morphology, and there appear to be different cell types (different morphology and size, including nuclei) compared to the other images. Could these be contaminating non-parenchymal cells (NPC)? Do they usually isolate only hepatocytes or are there any contaminating NPCs with their procedure?
- Out of curiosity: how long can they keep hepatocytes in culture before they begin to differentiate/lose the characteristics of hepatocytes?
This reviewer would like to compliment the authors on the detailed protocol and the advice/suggestions. By following this protocol, with the addition of the requested clarifications, even those without much experience can learn this difficult technique, which undoubtedly requires a lot of practice to be mastered.
Author Response
Responses to comments from Reviewer 3
Comment 1:
COLLAGENASE: they use 0,04% (wt/v) or 400 µg/ml. How many units does it correspond to?
Response
0.04% (wt/v) is equal to 400 µg/ml. 0.04% (wt/v) means 0.04 grams per 100 ml, which then can be converted to µg: 40000 µg /100 ml, and then converted to per mL: 400 µg/ml.
The specific activity as reported at the site of Sigma-Aldrich is 125 to 250 Mandl units per milligram of powdered substance.
Is it always the same or have they noticed differences between the various batches? Yes, the same.
Have you tried other brands besides the one indicated? NO, we used only Sigma-Aldrich brand.
Comment 2:
In Methods section, culture plates paragraph: It is useful they specify the total amount of collagen per cm2, thus allowing to repeat the process also for smaller or larger surface areas. They specify the collagen Type I and they write “(1:10)”. Does this mean they diluted the collagen 1:10? In what, PBS?
Response
We have already specified the concentration of collagen/cm2 which is equal to 10 µg/cm2 in line 156, page 4.
Yes, it was diluted in PBS. We have added this information and highlighted it in line 156, page 4.
Comment 3
The product they indicate is concentrated 6 mg/ml according to the Sigma-Aldrich website. If they dilute it 1:10, do they use it 0.6mg/ml? It is just to understand whether they use approximately 170 µl/well in a 6 well plate or they perform an intermediate dilution.
Response
According to the Sigma-Aldrich website for the product collagen type I (C4243) https://www.sigmaaldrich.com/JO/en/product/sigma/c4243?srsltid=AfmBOormngzV38QYgmwJBWedkX3JAbxh27GY5Zr_qbSJgZzNgDEp2UcG, the concentration is 2.9-3.2 mg/mL. The product datasheet documents the concentration as approximately 3 mg/mL. Given that the surface area of a well in a 6-well plate is 9.6 cm² and the desired coating concentration is 10 µg/cm², the required coating volume is 320 µL of 1:10 diluted collagen solution per well.
Comment 4:
In the Hepatocyte Dissociation and Purification section, they use a magnetic stirrer for digested liver dissociation. Does the size and the shape of the stirrer bar influence the outcome (e.g. cylindrical or pivot ring stirring bar)? What is the meaning of “gentle stirring”? The minimum possible?
Response
We used a pivot ring stirring bar with a length of 3 cm and did not test other types of stirring bars to compare their effects on dissociation. We have added this information as highlighted text in lines 208-209, page 6.
The stirring speed was approximately 800 rpm using a heating magnetic stirrer (Velp scientifica, Arec.x, Italy). We have added this specific information as highlighted text in lines 208-209, page 6.
Comment 5:
What is the diameter and the length of tubing they use? Does the tubing length and diameter influence the outcome of the perfusion?
Response
It is 1.5 m in length and 3 mm in diameter. We have added these details as highlighted text in lines 163-164, page 4.
It mainly affects the flow rate and ultimately the perfusion outcome.
Comment 6:
Can this procedure be used to isolate murine hepatocytes? Obviously the flow rate need adjustments, but did the authors use this procedure for mouse livers?
Response
This protocol can be used for mice with some adjustments to accommodate the smaller size of the animal, liver tissue, and blood vessels. For example, adjustments may include a reduced flow rate (e.g., 3 mL/min), shorter perfusion time, smaller cannula size (26G), and the use of a small stirring bar during hepatocyte dissociation. We have added this information as highlighted text in lines 518-526, page 15.
No, we actually did not use this procedure for mouse.
Comment 7:
In Discussion, lines 366-367: please specify that are rat hepatocytes.
Response
We have added and highlighted the term “rat hepatocytes” in a specific sentence on lines 370-371, page 12. Please note that the line numbers have shifted after editing while responding to the reviewers’ comments.
Comment 8:
In figure 7 A and B: The hepatocytes have a slightly different morphology, and there appear to be different cell types (different morphology and size, including nuclei) compared to the other images. Could these be contaminating non-parenchymal cells (NPC)? Do they usually isolate only hepatocytes or are there any contaminating NPCs with their procedure?
Response
Our protocol is primarily designed to isolate hepatocytes, which typically display a distinct morphology compared to non-parenchymal cells (NPCs) in culture. While the method enriches for hepatocytes, small numbers of NPCs can sometimes be present as contaminants. Figure 7 (A and B) shows normal hepatocytes cultured in standard media and they exhibit the same hepatic morphology.
Comment 9:
Out of curiosity: how long can they keep hepatocytes in culture before they begin to differentiate/lose the characteristics of hepatocytes?
Response
We maintained primary rat hepatocytes in culture for 92 hrs., during which they remained viable and retained their morphology. However, we did not specifically assess their functional activity or expression of hepatic markers at that time.
Comment 10
This reviewer would like to compliment the authors on the detailed protocol and the advice/suggestions. By following this protocol, with the addition of the requested clarifications, even those without much experience can learn this difficult technique, which undoubtedly requires a lot of practice to be mastered.
Response
We appreciate your complements. We are hoping that other researchers benefit from sharing our experience with the scientific community.
Round 2
Reviewer 1 Report
Comments and Suggestions for Authors
The authors have addressed some of the points raised in the review, particularly those requiring clarifications (e.g., providing the n-value). However, the responses to the more significant, methodological criticisms are largely defensive and rely on textual changes or citations to external literature rather than providing new experimental evidence.
- About major comment 1: The response to comment 1 does not address the core of the critique. A detailed description of parameters is not equivalent to objective evidence of superiority.
- About major comment 2: While the cited literature is valid, my point was to demonstrate its utility within the specific context of this hepatocyte isolation protocol. If the corresponding experiment is not feasible, the authors should at least modify the manuscript text to clarify that the inclusion of NAC is based onestablished literature from other fields, rather than presenting it as a factor proven to be effective herein.
- About Minor Comment 1: A manuscript must stand on its own merits based on the data presented within it. Data from an ongoing study are not available for review and therefore cannot be used to support the claims of the current manuscript. The language in the abstract must be revised to reflect the actual content. For example, it could state that the protocol "establishes a model" or "provides a foundation for studying" fatty liver disease.
- About Minor Comment 2: It is better to re-edit the introduction to remove the specific mention of STIM1 and TRPV1. The introduction should focus on the general need for high-quality hepatocytes to study NAFLD/NASH pathophysiology, making the manuscript a cohesive whole without references to unrelated work.
Author Response
Response to Reviewer 1
We thank the reviewer for their comments and efforts to improve the manuscript. We have revised the text accordingly and provided a detailed response to each comment.
- About major comment 1: The response to comment 1 does not address the core of the critique. A detailed description of parameters is not equivalent to objective evidence of superiority.
Response:
- In response to this comment, comparative data demonstrating superiority over a "standard" or commonly used protocol in terms of yield and viability have been provided. The paragraph added during the first-round revision (lines 380–390, pages 12–13) has been re-stated and is now highlighted in green as follows:
“We tested several previously established protocols employing the conventional two-step collagenase perfusion method for primary hepatocyte isolation. Although some of these protocols reported high yields and cell viability, their results were not reproducible, and the number of replicates was often not specified. For example, Shen et al. (2012) [2] reported isolating approximately 1.0 × 10⁸ cells from a single rat liver with viability ranging from 88% to 96%; however, we were unable to reproduce these outcomes in our laboratory using the same procedure. Similarly, Salem et al. (2018) [8] reported isolating 20 × 10⁶ total cells per mouse liver, but this protocol also proved irreproducible in our hands. Other studies have reported comparable yields for mouse liver [9,12], yet they likewise suffer from limited replication. A critical limitation across these reports is the insufficient number of replicate isolations, which undermines the strength of their reports regarding reproducibility. In contrast, our protocol consistently yields 75-90 million viable hepatocytes per rat liver, with cell viability ranging from 86% to 93% in each replicate, thereby demonstrating superior reliability and reproducibility compared to earlier methods”. Lines 371-384, pages 12-13.
- About major comment 2: While the cited literature is valid, my point was to demonstrate its utility within the specific context of this hepatocyte isolation protocol. If the corresponding experiment is not feasible, the authors should at least modify the manuscript text to clarify that the inclusion of NAC is based on established literature from other fields, rather than presenting it as a factor proven to be effective herein.
Response:
- We have revised the relevant section of the discussion according to your suggestion. The sentence in the manuscript (lines 395–398, page 13) has been revised and is now highlighted in green as follows:
“Therefore, we found that maintaining the buffer sterile, warm, oxygenated, and at a neutral pH is effective in keeping the cells viable and healthy during perfusion. Furthermore, the buffer was supplemented with N-acetyl-L-cysteine, which has been previously reported to have hepatoprotective properties, prevent oxidative damage, and improve cell viability in various cell types [21–25].” Line 393-398, page 13. The relevant references have been added, and the reference list has been updated accordingly.
- Yedjou CG, Tchounwou PB. N-acetyl-l-cysteine affords protection against lead-induced cytotoxicity and oxidative stress in human liver carcinoma (HepG2) cells. J. Environ. Res. Public health. 2007;4(2):132-7.
- Partyka A, Niżański W, Bratkowska M, Maślikowski P. Effects of N-acetyl-L-cysteine and catalase on the viability and motility of chicken sperm during liquid storage. Biol. 2015;15(2):126-9.
- Hosseini E, Ghasemzadeh M, Atashibarg M, Haghshenas M. ROS scavenger, N‐acetyl‐l‐cysteine and NOX specific inhibitor, VAS2870 reduce platelets apoptosis while enhancing their viability during storage. Transfusion. 2019;59(4):1333-43.
- About Minor Comment 1: A manuscript must stand on its own merits based on the data presented within it. Data from an ongoing study are not available for review and therefore cannot be used to support the claims of the current manuscript. The language in the abstract must be revised to reflect the actual content. For example, it could state that the protocol "establishes a model" or "provides a foundation for studying" fatty liver disease.
Response:
- We have revised the abstract according to your suggestion. The sentence in the abstract of the manuscript (Lines 21-22, page 1) has been revised and is now highlighted in green as:
“It also establishes an in vitro steatosis model for evaluating therapeutic drugs and molecular interventions.” Lines 21-22, page 1.
- The sentence in the abstract of the manuscript (Lines 33-35, page 1) has been revised and is now highlighted in green as:
“Additionally, the protocol provides a foundation for studying the pathophysiology of fatty liver disease.” Lines 33-34, page 1.
- About Minor Comment 2: It is better to re-edit the introduction to remove the specific mention of STIM1 and TRPV1. The introduction should focus on the general need for high-quality hepatocytes to study NAFLD/NASH pathophysiology, making the manuscript a cohesive whole without references to unrelated work.
Response:
- We have revised the introduction and conclusion according to your suggestion. The following paragraph from the previous version of the manuscript:
“Notably, isolated hepatocytes provide a good tool to test the functionality of receptors and channels, such as store-operated Ca2+ channels, stromal interaction molecule 1 (STIM1), transient receptor potential vanilloid 1 (TRPV1), and receptor-coupled phospholipase C, as well as the underlying signaling pathways using electrophysiology, patch clamp, or calcium imaging techniques [6-8].” Lines 46-50, page 2, has been deleted along with its corresponding references [6-8] from the references list. The remaining references have been re-numbered accordingly to reflect this change.
- Additionally, we removed the TRPV1 from the last sentence of the conclusion in the previous version of the manuscript:
“In this work, we provide guidelines for the successful harvesting and culturing of large number of functional hepatocyte as well as inducing steatosis in these cells as a prerequisite for the study of lipid metabolism and receptors (e.g., TRPV1)” in lines 611-614, page 17.
This sentence now reads:
“In this work, we provide guidelines for the successful harvesting and culturing of large number of functional hepatocyte as well as inducing steatosis in these cells as a prerequisite for the study of lipid metabolism.” Lines 611-614, page 17.
Round 3
Reviewer 1 Report
Comments and Suggestions for Authors
The problems have been addressed by the authors.